# APPROXIMATING HOW SINGLE HEAD ATTENTION LEARNS

## ABSTRACT

Why do models often attend to salient words, and how does this evolve throughout training? We approximate model training as a two stage process: early on in training when the attention weights are uniform, the model learns to translate individual input word $i$ to $o$ if they co-occur frequently. Later, the model learns to attend to $i$ while the correct output is $o$ because it knows $i$ translates to $o$. To formalize, we define a model property, Knowledge to Translate Individual Words (KTIW) (e.g. knowing that $i$ translates to $o$), and claim that it drives the learning of the attention. This claim is supported by the fact that before the attention mechanism is learned, KTIW can be learned from word co-occurrence statistics, but not the other way around. Particularly, we can construct a training distribution that makes KTIW hard to learn, the learning of the attention fails, and the model cannot even learn the simple task of copying the input words to the output. Our approximation explains why models sometimes attend to salient words, and inspires a toy example where a multi-head attention model can overcome the above hard training distribution by improving learning dynamics rather than expressiveness. We end by discussing the limitation of our approximation framework and suggest future directions.

## 1 INTRODUCTION

The attention mechanism underlies many recent advances in natural language processing, such as machine translation Bahdanau et al. (2015) and pretraining Devlin et al. (2019). While many works focus on analyzing attention in already-trained models Jain & Wallace (2019); Vashishth et al. (2019); Brunner et al. (2019); Elhage et al. (2021); Olsson et al. (2022), little is understood about how the attention mechanism is *learned* via gradient descent at training time.

These learning dynamics are important, as standard, gradient-trained models can have very unique inductive biases, distinguishing them from more esoteric but equally accurate models. For example, in text classification, while standard models typically attend to salient (high gradient influence) words Serrano & Smith (2019), recent work constructs accurate models that attend to irrelevant words instead Wiegreffe & Pinter (2019); Pruthi et al. (2020). In machine translation, while the standard gradient descent cannot train a high-accuracy transformer with relatively few attention heads, we can construct one by first training with more heads and then pruning the redundant heads Voita et al. (2019); Michel et al. (2019). To explain these differences, we need to understand how attention is learned at training time.

Our work opens the black box of attention training, focusing on attention in LSTM Seq2Seq models Luong et al. (2015) (Section 2.1). Intuitively, if the model knows that the input individual word $i$ translates to the correct output word $o$, it should attend to $i$ to minimize the loss. This motivates us to investigate the model's knowledge to translate individual words (abbreviated as KTIW), and we define a lexical probe $\beta$ to measure this property.

We claim that KTIW drives the attention mechanism to be learned. This is supported by the fact that KTIW can be learned when the attention mechanism has not been learned (Section 3.2), but not the other way around (Section 3.3). Specifically, even when the attention weights are frozen to be uniform, probe $\beta$ still strongly agrees with the attention weights of a standardly trained model. On the other hand, when KTIW cannot be learned, the attention mechanism cannot be learned. Particularly, we can construct a distribution where KTIW is hard to learn; as a result, the model fails to learn a simple task of copying the input to the output.

Now the problem of understanding how attention mechanism is learned reduces to understanding how KTIW is learned. Section 2.3 builds a simpler proxy model that approximates how KTIW is learned, and Section 3.2 verifies empirically that the approximation is reasonable. This proxy model is simple enough to analyze and we interpret its training dynamics with the classical IBM Translation Model 1 (Section 4.2), which translates individual word $i$ to $o$ if they co-occur more frequently.

To collapse this chain of reasoning, we approximate model training in two stages. Early on in training when the attention mechanism has not been learned, the model learns KTIW through word co-occurrence statistics; KTIW later drives the learning of the attention.

Using these insights, we explain why attention weights sometimes correlate with word saliency in binary text classification (Section 5.1): the model first learns to "translate" salient words into labels, and then attend to them. We also present a toy experiment (Section 5.2) where multi-head attention improves learning dynamics by combining differently initialized attention heads, even though a single head model can express the target function.

Nevertheless, "all models are wrong". Even though our framework successfully explains and predicts the above empirical phenomena, it cannot fully explain the behavior of attention-based models, since approximations are after all less accurate. Section 6 identifies and discusses two key assumptions: (1) information of a word tends to stay in the local hidden state (Section 6.1) and (2) attention weights are free variables (Section 6.2). We discuss future directions in Section 7.

## 2 MODEL

Section 2.1 defines the LSTM with attention Seq2Seq architecture. Section 2.2 defines the lexical probe $\beta$, which measures the model's knowledge to translate individual words (KTIW). Section 2.3 approximates how KTIW is learned early on in training by building a "bag of words" proxy model. Section 2.4 shows that our framework generalizes to binary classification.

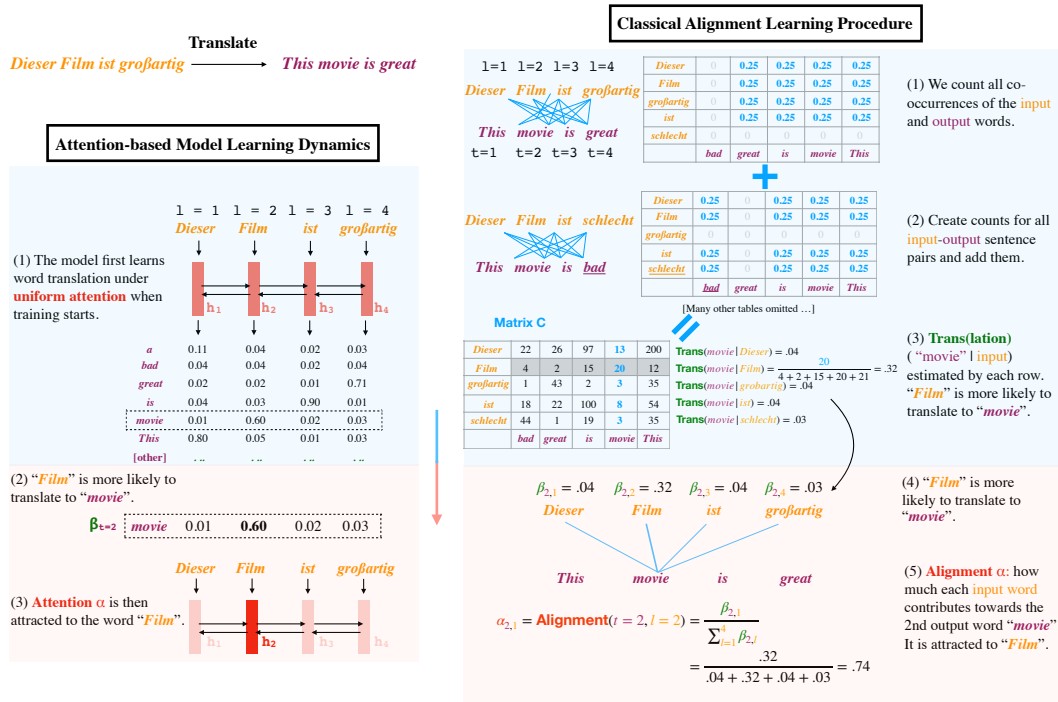

Figure 1: Attention mechanism in recurrent models (left, Section 2.1) and word alignments in the classical model (right, Section 4.2) are *learned* similarly. Both first learn how to translate individual words (KTIW) under uniform attention weights/alignment at the start of training (upper, blue background), which then drives the attention mechanism/alignment to be learned (lower, red background).

## 2.1 MACHINE TRANSLATION MODEL

We use the dot-attention variant from Luong et al. (2015). The model maps from an input sequence $\{x_l\}$ with length $L$ to an output sequence $\{y_t\}$ with length $T$. We first use LSTM encoders to embed $\{x_l\} \subset \mathcal{I}$ and $\{y_t\} \subset \mathcal{O}$ respectively, where $\mathcal{I}$ and $\mathcal{O}$ are input and output vocab space, and obtain encoder and decoder hidden states $\{h_l\}$ and $\{s_t\}$. Then we calculate the attention logits $a_{t,l}$ by applying a learnable mapping from $h_l$ and $s_t$, and use softmax to obtain the attention weights $\alpha_{t,l}$:

$$a_{t,l} = s_t^T W h_l; \quad \alpha_{t,l} = \frac{e^{a_{t,l}}}{\sum_{l'=1}^{L} e^{a_{t,l'}}}. \tag{1}$$

Next we sum the encoder hidden states $\{h_t\}$ weighted by the attention to obtain the "context vector" $c_t$, concatenate it with the decoder $s_t$, and obtain the output vocab probabilities $p_t$ by applying a learnable neural network $N$ with one hidden layer and softmax activation at the output, and train the model by minimizing the sum of negative log-likelihood of all the output words $y_t$.

$$c_t = \sum_{l=1}^{L} \alpha_{t,l} h_l; \quad p_t = N([c_t, s_t]); \quad \mathcal{L} = -\sum_{t=1}^{T} \log p_{t,y_t}. \tag{2}$$

## 2.2 LEXICAL PROBE $\beta$

We define the lexical probe $\beta_{t,l}$ as:

$$\beta_{t,l} := N([h_l, s_t])_{y_t}, \tag{3}$$

which means "the probability assigned to the correct word $y_t$, if the network attends only to the input encoder state $h_l$". If we assume that $h_l$ only contains information about $x_l$, $\beta$ closely reflects KTIW, since $\beta$ can be interpreted as "the probability that $x_l$ is translated to the output $y_t$".

Heuristically, to minimize the loss, the attention weights $\alpha$ should be attracted to positions with larger $\beta_{t,l}$. Hence, we expect the learning of the attention to be driven by KTIW (Figure 1 left). We then discuss how KTIW is learned.

## 2.3 EARLY DYNAMICS OF LEXICAL KNOWLEDGE

To approximate how KTIW is learned early on in training, we build a proxy model by making a few simplifying assumptions. First, since attention weights are uniform early on in training, we replace the attention distribution with a uniform one. Second, since we are defining individual word translation, we assume that information about each word is localized to its corresponding hidden state. Therefore, similar to Sun & Lu (2020), we replace $h_l$ with an input word embedding $e_{x_l} \in \mathbb{R}^d$, where $e$ represents the word embedding matrix and $d$ is the embedding dimension. Third, to simplify analysis, we assume $N$ only contains one linear layer $W \in \mathbb{R}^{|\mathcal{O}| \times d}$ before softmax activation and ignore the decoder state $s_t$. Putting these assumptions together, we now define a new proxy model that produces output vocab probability $p_t := \sigma(\frac{1}{L} \sum_{l=1}^{L} W e_{x_l})$.

On a high level, this proxy averages the embeddings of the input "bag of words", and produces a distribution over output vocabs to predict the output "bag of words". This implies that the sets of input and output words for each sentence pair are sufficient statistics for this proxy. The probe $\beta^{\text{px}}$ can be similarly defined as $\beta_{t,l}^{\text{px}} := \sigma(W e_{x_l})_{y_t}$.

We provide more intuitions on how this proxy learns in Section 4.

## 2.4 BINARY CLASSIFICATION MODEL

Binary classification can be reduced to "machine translation", where $T = 1$ and $|\mathcal{O}| = 2$. We drop the subscript $t = 1$ when discussing classification.

We use the standard architecture from Wiegreffe & Pinter (2019). After obtaining the encoder hidden states $\{h_t\}$, we calculate the attention logits $a_l$ by applying a feed-forward neural network with one hidden layer and take the softmax of $a$ to obtain the attention weights $\alpha$:

$$a_l = v^T(ReLU(Qh_l)); \quad \alpha_l = \frac{e^{a_l}}{\sum_{l'=1}^{L} e^{a_{l'}}} \quad, \tag{4}$$

where $Q$ and $v$ are learnable.

We sum the hidden states $\{h_l\}$ weighted by the attention, feed it to a final linear layer and apply the sigmoid activation function ($\sigma$) to obtain the probability for the positive class

$$p^{\text{pos}} = \sigma(W^T \sum_{l=1}^{L} a_l h_l) = \sigma(\sum_{l=1}^{L} \alpha_l W^T h_l). \tag{5}$$

Similar to the machine translation model (Section 2.1), we define the "lexical probe":

$$\beta_l := \sigma((2y - 1)W^T h_l), \tag{6}$$

where $y \in \{0, 1\}$ is the label and $2y - 1 \in \{-1, 1\}$ controls the sign.

On a high level, Sun & Lu (2020) focuses on binary classification and provides almost the exact same arguments as ours. Specifically, their polarity score "$s_l$" equals $\frac{\beta_l}{1 - \beta_l}$ in our context, and they provide a more subtle analysis of how the attention mechanism is learned in binary classification.

## 3 EMPIRICAL EVIDENCE

We provide evidence that KTIW drives the learning of the attention early on in training: KTIW can be learned when the attention mechanism has not been learned (Section 3.2), but not the other way around (Section 3.3).

### 3.1 MEASURING AGREEMENT

We start by describing how to evaluate the agreement between quantities of interest, such as $\alpha$ and $\beta$. For any input-output sentence pair $(x^m, y^m)$, for each output index $t$, $\alpha_t^m, \beta_t^m, \beta_t^{\text{px},m} \in \mathbb{R}^{L^m}$ all associate each input position $l$ with a real number. Since attention weights and word alignment tend to be sparse, we focus on the agreement of the highest-valued position. Suppose $u, v \in \mathbb{R}^L$, we formally define the agreement of $v$ with $u$ as:

$$\mathcal{A}(u, v) := \mathbf{1}[|\{j|v_j > v_{\arg\max u_i}\}| < 5\%L], \tag{7}$$

which means "whether the highest-valued position (dimension) in $u$ is in the top 5% highest-valued positions in $v$". We average the $\mathcal{A}$ values across all output words on the validation set to measure the agreement between two model properties. We also report Kendall's $\tau$ rank correlation coefficient in Appendix 2 for completeness.

We denote its random baseline as $\hat{\mathcal{A}}$. $\hat{\mathcal{A}}$ is close to but not exactly $5\%$ because of integer rounding.

**Contextualized Agreement Metric.** However, since different datasets have different sentence length distributions and variance of attention weights caused by random seeds, it might be hard to directly interpret this agreement metric. Therefore, we contextualize this metric with model performance. We use the standard method to train a model till convergence using $\mathcal{T}$ steps and denote its attention weights as $\alpha$; next we train the same model from scratch again using another random seed. We denote its attention weights at training step $\tau$ as $\hat{\alpha}(\tau)$ and its performance as $\hat{p}(\tau)$. Roughly speaking, when $\tau < \mathcal{T}$, both $\mathcal{A}(\alpha, \hat{\alpha}(\tau))$ and $\hat{p}(\tau)$ increase as $\tau$ increases. We define the contextualized agreement $\xi$ as:

$$\xi(u, v) := \hat{p}(\inf\{\tau | \mathcal{A}(\alpha, \hat{\alpha}(\tau)) > \mathcal{A}(u, v)\}). \tag{8}$$

In other words, we find the training step $\tau_0$ where its attention weights $\hat{\alpha}(\tau_0)$ and the standard attention weights $\alpha$ agrees more than $u$ and $v$ agrees, and report the performance at this iteration. We refer to the model performance when training finishes ($\tau = \mathcal{T}$) as $\xi^*$.

**Datasets.** We evaluate the agreement metrics $\mathcal{A}$ and $\xi$ on multiple machine translation and text classification datasets. For machine translation, we use Multi-30k (En-De), IWSLT'14 (De-En), and News Commentary v14 (En-Nl, En-Pt, and It-Pt). For text classification, we use IMDB Sentiment Analysis, AG News Corpus, 20 Newsgroups (20 NG), Stanford Sentiment Treebank, Amazon review,

| Task | $\mathcal{A}(\alpha, \beta^{\mathrm{uf}})$ | $\mathcal{A}(\beta^{\mathrm{uf}}, \beta^{\mathrm{px}})$ | $\mathcal{A}(\Delta, \beta^{\mathrm{uf}})$ | $\mathcal{A}(\alpha, \beta)$ | $\hat{\mathcal{A}}$ | $\xi(\alpha, \beta^{\mathrm{uf}})$ | $\xi(\alpha, \beta)$ | $\xi^*$ |
|---|---|---|---|---|---|---|---|---|
| IMDB | 53 | 82 | 62 | 60 | 5 | 87 | 87 | 90 |
| AG News | 39 | 55 | 43 | 48 | 6 | 94 | 95 | 96 |
| 20 NG | 65 | 41 | 65 | 63 | 5 | 91 | 85 | 94 |
| SST | 20 | 34 | 22 | 25 | 8 | 78 | 82 | 84 |
| Multi30k | 31 | 34 | 27 | 49 | 7 | 43 | 49 | 66 |
| IWSLT14 | 36 | 39 | 28 | 55 | 7 | 36 | 44 | 67 |
| News It-Pt | 29 | 39 | 25 | 52 | 6 | 22 | 25 | 55 |

Table 1: The tasks above the horizontal line are classification and below are translation. The (contextualized) agreement metric $\mathcal{A}(\xi)$ is described in Section 3.1. Across all tasks, $\mathcal{A}(\alpha, \beta)$, $\mathcal{A}(\alpha, \beta^{\mathrm{uf}})$, and $\mathcal{A}(\beta^{\mathrm{uf}}, \beta^{\mathrm{px}})$ significantly outperform the random baseline $\hat{\mathcal{A}}$ and the corresponding contextualized interpretations $\xi$ are also non-trivial. This implies that 1) the proxy model from Section 2.3 approximates well how KTIW is learned, 2) attention weights $\alpha$ and the probe $\beta$ of KTIW strongly agrees, and 3) KTIW can still be learned when the attention weights are uniform.

and Yelp Open Data Set. All of them are in English. The details and citations of these datasets can be seen in the Appendix A.5. We use token accuracy[1] to evaluate the performance of translation models and accuracy to evaluate the classification models.

Due to space limit we round to integers and include a subset of datasets in Table 1 for the main paper. Appendix Table 4 includes the full results.

## 3.2 KTIW LEARNS UNDER UNIFORM ATTENTION

Even when the attention mechanism has not been learned, KTIW can still be learned. We train the same model architecture with the attention weights frozen to be *uniform*, and denote its lexical probe as $\beta^{\mathrm{uf}}$. Across all tasks, $\mathcal{A}(\alpha, \beta^{\mathrm{uf}})$ and $\mathcal{A}(\beta^{\mathrm{uf}}, \beta^{\mathrm{px}})$ [2] significantly outperform the random baseline $\hat{\mathcal{A}}$, and the contextualized agreement $\xi(\alpha, \beta^{\mathrm{uf}})$ is also non-trivial. This indicates that 1) the proxy we built in Section 2.3 approximates KTIW and 2) even when the attention weights are uniform, KTIW is still learned.

## 3.3 ATTENTION FAILS WHEN KTIW FAILS

We consider a simple task of copying from the input to the output, and each input is a permutation of the same set of 40 vocab types. Under this training distribution, the proxy model provably cannot learn: every input-output pair contains the exact same set of input-output words.[3] As a result, our framework predicts that KTIW is unlikely to be learned, and hence the learning of attention is likely to fail.

The training curves of learning to copy the permutations are in Figure 2 left, colored in red: the model sometimes fails to learn. For the control experiment, if we randomly sample and permute 40 vocabs from 60 vocab types as training samples, the model successfully learns (blue curve) from this distribution every time. Therefore, even if the model is able to express this task, it might fail to learn it when KTIW is not learned. The same qualitative conclusion holds for the training distribution that mixes permutations of two disjoint sets of words (Figure 2 middle), and Appendix A.3 illustrates the intuition.

For binary classification, it follows from the model definition that attention mechanism cannot be learned if KTIW cannot be learned, since

$$p^{\mathrm{correct}} = \sigma(\sum_{l=1}^{L} \alpha_l \sigma^{-1}(\beta_l)); \quad \sigma(x) = \frac{1}{1 + e^{-x}}, \tag{9}$$

---

[1]Appendix Tables 5, 3, and 7 include results for BLEU.

[2]Empirically, $\beta^{\mathrm{px}}$ converges to the unigram weight of a bag-of-words logistic regression model, and hence $\beta^{\mathrm{px}}$ does capture an interpretable notion of "keywords". (Appendix A.10.)

[3]We provide more intuitions on this in Section 4

and the model needs to attend to positions with higher $\beta$, in order to predict correctly and minimize the loss. For completeness, we include results where we freeze $\beta$ and find that the learning of the attention fails in Appendix A.6.

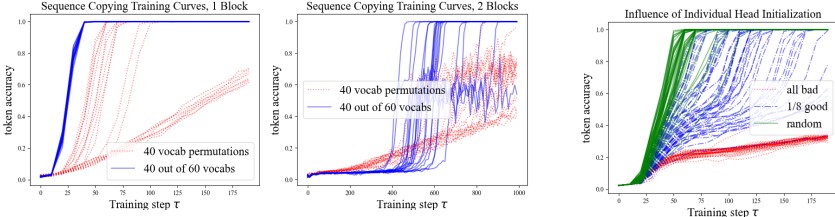

Figure 2: Each curve represents accuracy on the test distribution vs. number of training steps for different random seeds (20 each). Left and Middle are accuracy curves for single head attention models. When trained on a distribution of permutation of 40 vocabs (red) (Left) or a mixture of permutations (Middle), the model sometimes fails to learn and converges slower. The right figure is for multi-head attention experiments. If all head initializations (head-init) are bad (red), the model is likely to fail; if one of the head-init is good (blue), it is likely to learn; with high chance, at least one out of eight random head-init is good (green).

## 4 CONNECTION TO IBM MODEL 1

Section 2.3 built a simple proxy model to approximate how KTIW is learned when the attention weights are uniform early on in training, and Section 3.2 verified that such an approximation is empirically sound. However, it is still hard to intuitively reason about how this proxy model learns. This section provides more intuitions by connecting its initial gradient (Section 4.1) to the classical IBM Model 1 alignment algorithm Brown et al. (1993) (Section 4.2).

### 4.1 DERIVATIVE AT INITIALIZATION

We continue from the end of Section 2.3. For each input word $i$ and output word $o$, we are interested in understanding the probability that $i$ assigns to $o$, defined as:

$$\theta_{i,o}^{\mathrm{px}} := \sigma(We_i)_o. \tag{10}$$

This quantity is directly tied to $\beta^{\mathrm{px}}$, since $\beta_{t,l}^{\mathrm{px}} = \theta_{x_l,y_t}^{\mathrm{px}}$. Using super-script $m$ to index sentence pairs in the dataset, the total loss $\mathcal{L}$ is:

$$\mathcal{L} = -\sum_m \sum_{t=1}^{T^m} \log(\sigma(\frac{1}{L^m} \sum_{l=1}^{L^m} We_{x_l^m})_{y_t^m}). \tag{11}$$

Suppose each $e_i$ or $W_o$ is independently initialized from a normal distribution $\mathcal{N}(0, I_d/d)$ and we minimize $\mathcal{L}$ over $W$ and $e$ using gradient flow, then the value of $e$ and $W$ are uniquely defined for each continuous time step $\tau$. By some straightforward but tedious calculations (details in Appendix A.2), the derivative of $\theta_{i,o}$ when the training starts is:

$$\lim_{d \to \infty} \frac{\partial \theta_{i,o}^{\mathrm{px}}}{\partial \tau}(\tau = 0) \xrightarrow{p} 2(C_{i,o}^{\mathrm{px}} - \frac{1}{|\mathcal{O}|} \sum_{o' \in \mathcal{O}} C_{i,o'}^{\mathrm{px}}). \tag{12}$$

where $\xrightarrow{p}$ means convergence in probability and $C_{i,o}^{\mathrm{px}}$ is defined as

$$C_{i,o}^{\mathrm{px}} := \sum_m \sum_{l=1}^{L^m} \sum_{t=1}^{T^m} \frac{1}{L^m} \mathbf{1}[x_l^m = i] \mathbf{1}[y_t^m = o]. \tag{13}$$

Equation 12 tells us that $\beta_{t,l}^{\mathrm{px}} = \theta_{x_l,y_t}^{\mathrm{px}}$ is likely to be larger if $C_{x_l,y_t}$ is large. The definition of $C$ seems hard to interpret from Equation 13, but in the next subsection we will find that this quantity naturally corresponds to the "count table" used in the classical IBM 1 alignment learning algorithm.

## 4.2 IBM MODEL 1 ALIGNMENT LEARNING

The classical alignment algorithm aims to learn which input word is responsible for each output word (e.g. knowing that $y_2$ "movie" aligns to $x_2$ "Film" in Figure 1 upper left), from a set of input-output sentence pairs. IBM Model 1 Brown et al. (1993) starts with a 2-dimensional count table $C^{\mathrm{IBM}}$ indexed by $i \in \mathcal{I}$ and $o \in \mathcal{O}$, denoting input and output vocabs. Whenever vocab $i$ and $o$ co-occurs in an input-output pair, we add $\frac{1}{L}$ to the $C^{\mathrm{IBM}}_{i,o}$ entry (step 1 and 2 in Figure 1 right). After updating $C^{\mathrm{IBM}}$ for the entire dataset, $C^{\mathrm{IBM}}$ is exactly the same as $C^{\mathrm{px}}$ defined in Equation 13. We drop the super-script of $C$ to keep the notation uncluttered.

Given $C$, the classical model estimates a probability distribution of "what output word $o$ does the input word $i$ translate to" (Figure 1 right step 3) as

$$\mathrm{Trans}(o|i) = \frac{C_{i,o}}{\sum_{o'} C_{i,o'}}. \tag{14}$$

In a pair of sequences $(\{x_l\}, \{y_t\})$, the probability $\beta^{\mathrm{IBM}}$ that $x_l$ is translated to the output $y_t$ is:

$$\beta^{\mathrm{IBM}}_{t,l} := \mathrm{Trans}(y_t|x_l), \tag{15}$$

and the alignment probability $\alpha^{\mathrm{IBM}}$ that "$x_l$ is responsible for outputting $y_t$ versus other $x_{l'}$" is

$$\alpha^{\mathrm{IBM}}(t,l) = \frac{\beta^{\mathrm{IBM}}_{t,l}}{\sum_{l'=1}^{L} \beta^{\mathrm{IBM}}_{t,l'}}, \tag{16}$$

which monotonically increases with respect to $\beta^{\mathrm{IBM}}_{t,l}$. See Figure 1 right step 5.

## 4.3 VISUALIZING AFOREMENTIONED TASKS

Figure 1 (right) visualizes the count table $C$ for the machine translation task, and illustrates how KTIW is learned and drives the learning of attention. We provide similar visualization for why KTIW is hard to learn under a distribution of vocab permutations (Section 3.3) in Figure 3, and how word polarity is learned in binary classification (Section 2.4) in Figure 4.

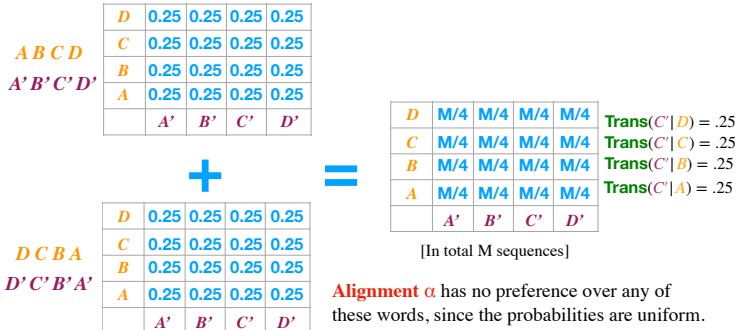

Figure 3: Co-occurrence table $C$ is non-informative under a distribution of permutations. Therefore, this distribution is hard for the attention-based model to learn.

# 5 APPLICATION

## 5.1 INTERPRETABILITY IN CLASSIFICATION

We use gradient based method Ebrahimi et al. (2018) to approximate the influence $\Delta_l$ for each input word $x_l$. The column $\mathcal{A}(\Delta, \beta^{\mathrm{uf}})$ reports the agreement between $\Delta$ and $\beta^{\mathrm{uf}}$, and it significantly outperforms the random baseline. Since KTIW initially drives the attention mechanism to be learned, this explains why attention weights are correlated with word saliency on many classification tasks, even though the training objective does not explicitly reward this.

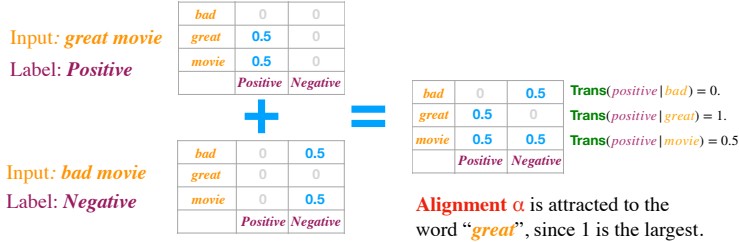

Figure 4: The classical model first learns word polarity, which later attracts attention.

## 5.2 MULTI-HEAD IMPROVES TRAINING DYNAMICS

We saw in Section 3.3 that learning to copy sequences under a distribution of permutations is hard and the model can fail to learn; however, sometimes it is still able to learn. Can we improve learning and overcome this hard distribution by ensembling several attention parameters together?

We introduce a multi-head attention architecture by summing the context vector $c_t$ obtained by each head. Suppose there are $K$ heads each indexed by $k$, similar to Section 2.1:

$$a_{t,l}^{(k)} = s_t^T W^{(k)} h_l; \quad \alpha_{t,l}^{(k)} = \frac{e^{a_{t,l}^{(k)}}}{\sum_{l'=1}^{L} e^{\alpha_{t,l'}^{(k)}}}, \tag{17}$$

and the context vector and final probability $p_t$ defined as:

$$c_t^{(k)} = \sum_{l=1}^{L} \alpha_{t,l}^{(k)} h_l; \quad p_t = N([\sum_{k=1}^{K} c_t^{(k)}, d_t]), \tag{18}$$

where $W^{(k)}$ are different learn-able parameters.

We call $W_{init}^{(k)}$ a good initialization if training with this single head converges, and bad otherwise. We use rejection sampling to find good/bad head initializations and combine them to form 8-head ($K = 8$) attention models. We experiment with 3 scenarios: (1) all head initializations are bad, (2) only one initialization is good, and (3) initializations are sampled independently at random.

Figure 2 right presents the training curves. If all head initializations are bad, the model fails to converge (red). However, as long as one of the eight initializations is good, the model can converge (blue). As the number of heads increases, the probability that all initializations are bad is exponentially small if all initializations are sampled independently; hence the model converges with very high probability (green). In this experiment, multi-head attention improves not by increasing expressiveness, since one head is sufficient to accomplish the task, but by improving the learning dynamics.

## 6 ASSUMPTIONS

We revisit the approximation assumptions used in our framework. Section 6.1 discusses whether the lexical probe $\beta_{t,l}$ necessarily reflects local information about input word $x_l$, and Section 6.2 discusses whether attention weights can be freely optimized to attend to large $\beta$. These assumptions are accurate enough to predict phenomenon in Section 3 and 5, but they are not *always* true and hence warrant more future researches. We provide simple examples where these assumptions might fail.

## 6.1 $\beta$ REMAINS LOCAL

We use a toy classification task to show that early on in training, expectantly, $\beta^{\text{uf}}$ is larger near positions that contain the keyword. However, unintuitively, $\beta_L^{\text{uf}}$ ($\beta$ at the last position in the sequence) will become the largest if we train the model for too long under uniform attention weights.

In this toy task, each input is a length-40 sequence of words sampled from $\{1, \dots, 40\}$ uniformly at random; a sequence is positive if and only if the keyword "1" appears in the sequence. We restrict "1"

to appear only once in each positive sequence, and use rejection sampling to balance positive and negative examples. Let $l^*$ be the position where $x_{l^*} = 1$.

For the positive sequences, we examine the log-odd ratio $\gamma_l$ before the sigmoid activation in Equation 5, since $\beta$ will be all close to 1 and comparing $\gamma$ would be more informative: $\gamma_l := \log \frac{\beta_l^{\text{uf}}}{1-\beta_l^{\text{uf}}}$.

We measure four quantities: 1) $\gamma_{l^*}$, the log-odd ratio if the model only attends to the key word position, 2) $\gamma_{l^*+1}$, one position after the key word position, 3) $\bar{\gamma} := \frac{\sum_{l=1}^{L} \gamma_l}{L}$, if attention weights are uniform, and 4) $\gamma_L$ if the model attends to the last hidden state. If the $\gamma_l$ only contains information about word $x_l$, we should expect:

$$\text{Hypothesis 1} : \gamma_{l^*} \gg \bar{\gamma} \gg \gamma_L \approx \gamma_{l^*+1}. \tag{19}$$

However, if we accept the conventional wisdom that hidden states contain information about nearby words Khandelwal et al. (2018), we should expect:

$$\text{Hypothesis 2} : \gamma_{l^*} \gg \gamma_{l^*+1} \gg \bar{\gamma} \approx \gamma_L. \tag{20}$$

To verify these hypotheses, we plot how $\gamma_{l^*}, \gamma_{l^*+1}, \bar{\gamma}$, and $\gamma_L$ evolve as training proceeds in Figure 5. Hypothesis 2 is indeed true when training starts; however, we find the following to be true asymptotically:

$$\text{Observation 3} : \gamma_L \gg \gamma_{l^*+1} \gg \bar{\gamma} \approx \gamma_{l^*}. \tag{21}$$

which is wildly different from Hypothesis 2. If we train under uniform attention weights for too long, the information about keywords can freely flow to other non-local hidden states.

## 6.2 Attention Weights are Free Variables

In Section 2.1 we assumed that attention weights $\alpha$ behave like free variables that can assign arbitrarily high probabilities to positions with larger $\beta$. However, $\alpha$ is produced by a model, and sometimes learning the correct $\alpha$ can be challenging.

Let $\pi$ be a random permutation of integers from 1 to 40, and we want to learn the function $f$ that permutes the input with $\pi$:

$$f([x_1, x_2, \ldots x_{40}]) := [x_{\pi(1)}, x_{\pi(2)} \ldots x_{\pi(40)}]. \tag{22}$$

Input $x$ are randomly sampled from a vocab of size 60 as in Section 3.3. Even though $\beta^{\text{uf}}$ behaves exactly the same for these two tasks, sequence copying is much easier to learn than permutation function: while the model always reaches perfect accuracy in the former setting within 300 iterations, it always fails in the latter.

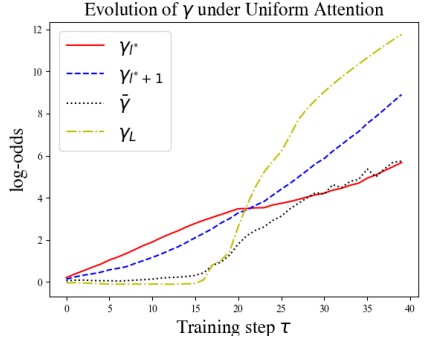

Figure 5: When training begins, Hypothesis 2 (Equation 19) is true; however, asymptotically, Oberservation 3 (Equation 21) is true.

LSTM has a built-in inductive bias to learn monotonic attention.

## 7 Conclusions

Our work tries to understand the black box of attention training. Early on in training, the LSTM attention models first learn how to translation individual words from bag of words co-occurrence statistics, which then drives the learning of the attention. Our framework explains why attention weights obtained by standard training often correlate with saliency, and how multi-head attention can increase performance by improving the training dynamics rather than expressiveness. These phenomena cannot be explained if we treated the training process as a black box.

## 8 Ethical Considerations

We present a new framework for understanding and predicting behaviors of an existing technology: the attention mechanism in recurrent neural networks. We do not propose any new technologies or any new datasets that could directly raise ethical questions. However, it is useful to keep in mind that our framework is far from solving the question of neural network interpretability, and should not be interpreted as ground truth in high stake domains like medicine or recidivism. We are aware and very explicit about the limitations of our framework, which we made clear in Section 6.

## 9 Reproducability Statement

To promote reproducibility, we provide extensive results in the appendix and describe all experiments in detail. We also attach source code for reproducing all experiments to the supplemental of this submission.

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

# A    APPENDICES

## A.1    HEURISTIC THAT $\alpha$ ATTENDS TO LARGER $\beta$

It is a heuristic rather than a rigorous theorem that attention $\alpha$ is attracted to larger $\beta$. There are two reasons. First, there is a non-linear layer after the averaging the hidden states, which can interact in an arbitrarily complex way to break this heuristic. Second, even if there are no non-linear operations after hidden state aggregation, the optimal attention that minimizes the loss does not necessarily assign *any* probability to the position with the largest $\beta$ value when there are more than two output vocabs.

Specifically, we consider the following model:

$$p_t = \sigma(W_c \sum_{l=1} \alpha_{t,l} h_l + W_s s_t) = \sigma(\sum_{l=1} \alpha_{t,l} \gamma_l + \gamma_s), \tag{23}$$

where $W_c$ and $W_s$ are learnable weights, and $\gamma$ defined as:

$$\gamma_l := W_c h_l; \quad \gamma_s := W_s s_t \Rightarrow \beta_{t,l} = \sigma(\gamma_l + \gamma_s)_{y_t}. \tag{24}$$

Consider the following scenario that outputs a probability distribution $p$ over 3 output vocabs and $\gamma_s$ is set to 0:

$$p = \sigma(\alpha_1 \gamma_1 + \alpha_2 \gamma_2 + \alpha_3 \gamma_3), \tag{25}$$

where $\gamma_{l=1,2,3} \in \mathbb{R}^{|\mathcal{O}|=3}$ are the logits, $\alpha$ is a valid attention probability distribution, $\sigma$ is the softmax, and $p$ is the probability distribution produced by this model. Suppose

$$\gamma_1 = [0, 0, 0], \gamma_2 = [0, -10, 5], \gamma_3 = [0, 5, -10] \tag{26}$$

and the correct output is the first output vocab (i.e. the first dimension). Therefore, we take the softmax of $\gamma_l$ and consider the first dimension:

$$\beta_{l=1} = \frac{1}{3} > \beta_{l=2} = \beta_{l=3} \approx e^{-5}. \tag{27}$$

We calculate "optimal $\alpha$" $\alpha^{\mathrm{opt}}$: the optimal attention weights that can maximize the correct output word probability $p_0$ and minimize the loss. We find that $\alpha_2^{\mathrm{opt}} = \alpha_3^{\mathrm{opt}} = 0.5$, while $\alpha_1^{\mathrm{opt}} = 0$. In this example, the optimal attention assigns 0 weight to the position $l$ with the highest $\beta_l$.

Fortunately, such pathological examples rarely occur in real datasets, and the optimal $\alpha$ are usually attracted to positions with higher $\beta$. We empirically verify this for the below variant of machine translation model on Multi30K.

As before, we obtain the context vector $c_t$. Instead of concatenating $c_t$ and $d_t$ and pass it into a non-linear neural network $N$, we add them and apply a linear layer with softmax after it to obtain the output word probability distribution

$$p_t = \sigma(W(c_t + d_t)). \tag{28}$$

This model is desirable because we can now provably find the optimal $\alpha$ using gradient descent (we delay the proof to the end of this subsection). Additionally, this model has comparable performance with the variant from our main paper (Section 2.1), achieving 38.2 BLEU score, vs. 37.9 for the model in our main paper. We use $\alpha^{\mathrm{opt}}$ to denote the attention that can minimize the loss, and we find that $\mathcal{A}(\alpha^{\mathrm{opt}}, \beta) = 0.53$. $\beta$ do strongly agree with $\alpha^{\mathrm{opt}}$.

Now we are left to show that we can use gradient descent to find the optimal attention weights to minimize the loss. We can rewrite $p_t$ as

$$p_t = \sigma(\sum_{l=1}^{L} \alpha_l W h_l + W d_t). \tag{29}$$

We define

$$\gamma_l := W h_l; \quad \gamma_s := W d_t. \tag{30}$$

Without loss of generality, suppose the first dimension of $\gamma_{1\ldots L}, \gamma_s$ are all 0, and the correct token we want to maximize probability for is the first dimension, then the loss for the output word is

$$\mathcal{L} = \log(1 + g(\alpha)), \tag{31}$$

where

$$g(\alpha) := \sum_{o \in \mathcal{O}, o \neq 0} e^{\alpha^T \gamma'_o + \gamma_{s,o}}, \tag{32}$$

where

$$\gamma'_o = [\gamma_{1,o} \ldots \gamma_{l,o} \ldots \gamma_{L,o}] \in \mathbb{R}^L. \tag{33}$$

Since $\alpha$ is defined within the convex probability simplex and $g(\alpha)$ is convex with respect to $\alpha$, the global optima $\alpha^{\mathrm{opt}}$ can be found by gradient descent.

## A.2 Calculating $\frac{\partial \theta_{i,o}}{\partial \tau}$

We drop the $^{\mathrm{px}}$ super-script of $\theta$ to keep the notation uncluttered. We copy the loss function here to remind the readers:

$$\mathcal{L} = -\sum_m \sum_{t=1}^{T^m} \log(\sigma(\frac{1}{L^m} \sum_{l=1}^{L^m} We_{x_l^m})_{y_t^m}). \tag{34}$$

and since we optimize $W$ and $e$ with gradient flow,

$$\frac{\partial W}{\partial \tau} := -\frac{\mathcal{L}}{\partial W}; \quad \frac{\partial e}{\partial \tau} := -\frac{\mathcal{L}}{\partial e}. \tag{35}$$

We first define the un-normalized logits $\hat{\gamma}$ and then take the softmax.

$$\hat{\theta} = We, \tag{36}$$

then

$$\frac{\partial \hat{\theta}}{\partial \tau} = \frac{\partial(We)}{\partial \tau} = -W\frac{\partial e}{\partial \tau} - \frac{\partial W}{\partial \tau}e. \tag{37}$$

We first analyze $\epsilon := -W\frac{\partial e}{\partial \tau}$. Since $\epsilon \in \mathbb{R}^{|\mathcal{I}| \times |\mathcal{O}|}$, we analyze each entry $\epsilon_{i,o}$. Since differentiation operation and left multiplication by matrix $W$ is linear, we analyze each individual loss term in Equation 34 and then sum them up.

We define

$$p^m := \sigma(\frac{1}{L^m} \sum_{l=1}^{L^m} We_{x_l^m}) \tag{38}$$

and

$$\mathcal{L}_t^m := -\log(p_{y_t^m}^m); \quad \epsilon_{t,i,o}^m := W_o \frac{\partial \mathcal{L}_t^m}{\partial e_i}. \tag{39}$$

Hence,

$$\mathcal{L} = \sum_m \sum_{t=1}^{T^m} \mathcal{L}_t^m; \quad \epsilon_{i,o} = \sum_m \sum_{t=1}^{T^m} \epsilon_{t,i,o}^m. \tag{40}$$

Therefore,

$$-\frac{\partial \mathcal{L}_t^m}{\partial e_i} = \frac{1}{L^m} \sum_{l=1}^{L^m} \mathbf{1}[x_l^m = i](W_{y_t^m} - \sum_{o=1}^{|\mathcal{O}|} p_o^m W_o). \tag{41}$$

Hence,

$$\epsilon_{t,i,y_t^m}^m = -W_{y_t^m}^T \frac{\partial \mathcal{L}_t^m}{\partial e_i} = \frac{1}{L^m} \sum_{l=1}^{L^m} \mathbf{1}[x_l^m = i] \tag{42}$$

$$(||W_{y_t^m}||_2^2 - \sum_{o=1}^{|\mathcal{O}|} p_o^m W_{y_t^m}^T W_o),$$

while for $o' \neq y_t^m$,

$$\epsilon_{t,i,o'}^m = -W_{o'}^T \frac{\partial \mathcal{L}_t^m}{\partial e_i} = \frac{1}{L^m} \sum_{l=1}^{L^m} \mathbf{1}[x_l^m = i] \tag{43}$$

$$(W_{o'}^T W_{y_t^m} - \sum_{o=1}^{|\mathcal{O}|} p_o^m W_{o'}^T W_o).$$

If $W_o$ and $e_i$ are each sampled i.i.d. from $\mathcal{N}(0, I_d/d)$, then by central limit theorem:

$$\forall o \neq o', \sqrt{d} W_o^T W_{o'} \xrightarrow{p} \mathcal{N}(0, 1), \tag{44}$$

$$\forall o, i, \sqrt{d} W_o^T e_i \xrightarrow{p} \mathcal{N}(0, 1), \tag{45}$$

and

$$\forall o, \sqrt{d}(||W_o||_2^2 - 1) \xrightarrow{p} \mathcal{N}(0, 2). \tag{46}$$

Therefore, when $\tau = 0$,

$$\lim_{d \to \infty} \epsilon_{t,i,o}^m \xrightarrow{p} \frac{1}{L^m} \sum_{l=1}^{L^m} \mathbf{1}[x_l^m = i](\mathbf{1}[y_l^t = o] - \frac{1}{|\mathcal{O}|}). \tag{47}$$

Summing over all the $\epsilon_{t,i,o}^m$ terms, we have that

$$\epsilon_{i,o} = C_{i,o} - \frac{1}{|\mathcal{O}|} \sum_{o'} C_{i,o'}, \tag{48}$$

where $C$ is defined as

$$C_{i,o} := \sum_m \sum_{l=1}^{L^m} \sum_{t=1}^{T^m} \frac{1}{L^m} \mathbf{1}[x_l^m = i] \mathbf{1}[y_t^m = o]. \tag{49}$$

We find that $-\frac{\partial W}{\partial \tau} e$ converges exactly to the same value. Hence

$$\frac{\partial \hat{\theta}_{i,o}}{\partial \tau} = \frac{\partial W e}{\partial \tau} = 2(C_{i,o} - \frac{1}{|\mathcal{O}|} \sum_{o'} C_{i,o'}). \tag{50}$$

Since $\lim_{d \to \infty} \theta(\tau = 0) \xrightarrow{p} \frac{1}{|\mathcal{O}|} \mathbf{1}^{|\mathcal{I}| \times |\mathcal{O}|}$, by chain rule,

$$\lim_{d \to \infty} \frac{\partial \gamma_{i,o}}{\partial \tau}(\tau = 0) \xrightarrow{p} 2(C_{i,o} - \frac{1}{|\mathcal{O}|} \sum_{o' \in \mathcal{O}} C_{i,o'}). \tag{51}$$

### A.3 MIXTURE OF PERMUTATIONS

For this experiment, each input is either a random permutation of the set $\{1 \ldots 40\}$, or a random permutation of the set $\{41 \ldots 80\}$. The proxy model can easily learn whether the input words are less than 40 and decide whether the output words are all less than 40. However, $\beta^{px}$ is still the same for every position; as a result, the attention and hence the model fail to learn. The count table $C$ can be see in Figure 6.

### A.4 ADDITIONAL TABLES FOR COMPLETENESS

We report several variants of Table 1. We chose to use token accuracy to contextualize the agreement metric in the main paper, because the errors would accumulate much more if we use a not-fully trained model to auto-regressively generate output words.

- Table 2 contains the same results as Table 1, except that its agreement score $\mathcal{A}(u, v)$ is now Kendall Tau rank correlation coefficient, which is a more popular metric.

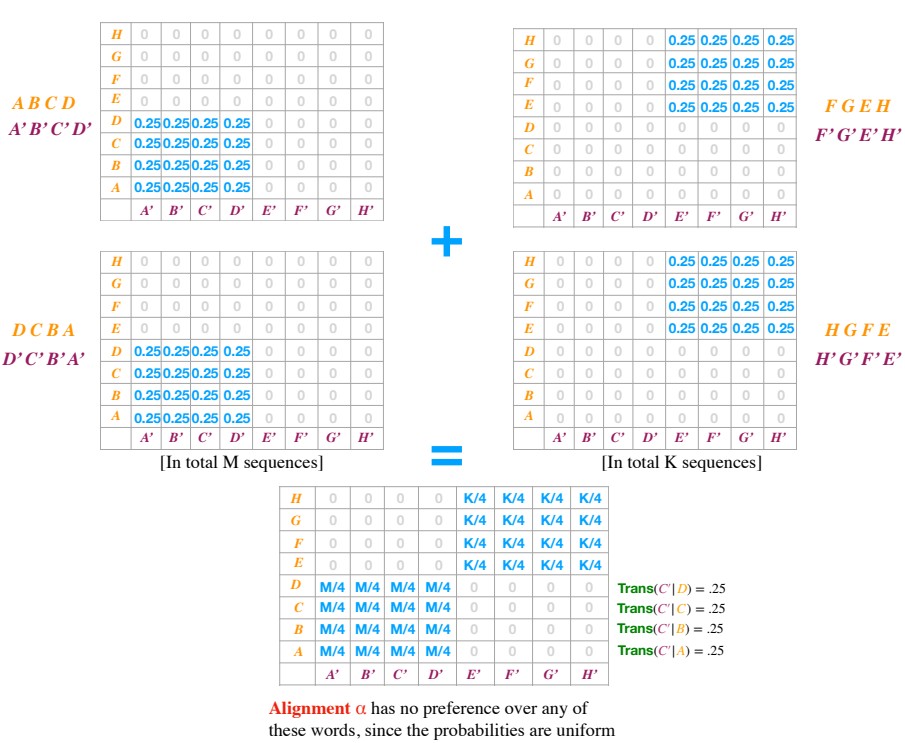

Figure 6: The training distributions mixes random permutation of disjoint set of words (left and right, respectively). From the count table, $\beta^{\text{px}}$ could learn that the set of input words $\{A, B, C, D\}$ corresponds to the set of output words $\{A', B', C', D'\}$, but its $\beta$ value for each input position is still uniformly 0.25.

| Task | $\mathcal{A}(\alpha, \beta^{\text{uf}})$ | $\mathcal{A}(\beta^{\text{uf}}, \beta^{\text{px}})$ | $\mathcal{A}(\Delta, \beta^{\text{uf}})$ | $\hat{\mathcal{A}}$ |
|---|---|---|---|---|
| IMDB | 12.77 | 33.31 | 12.56 | 0.00 |
| Yelp | 20.38 | 36.75 | 20.98 | 0.00 |
| AG News | 26.31 | 36.65 | 20.55 | 0.00 |
| 20 NG | 16.06 | 22.03 | 6.50 | 0.00 |
| SST | 11.68 | 31.43 | 15.01 | 0.00 |
| Amzn | 15.21 | 35.84 | 9.33 | 0.00 |
| Muti30k | 7.89 | 27.54 | 3.93 | 0.00 |
| IWSLT14 | 8.64 | 22.56 | 2.72 | 0.00 |
| News It-Pt | 4.82 | 17.16 | 1.63 | 0.00 |
| News En-Nl | 4.53 | 20.35 | 2.08 | 0.00 |
| News En-Pt | 4.65 | 18.20 | 2.17 | 0.00 |
| Task | $\xi(\alpha, \beta^{\text{uf}})$ | $\xi(\beta^{\text{uf}}, \beta^{\text{px}})$ | $\xi(\Delta, \beta^{\text{uf}})$ | $\xi^*$ |
| IMDB | 70.60 | 80.50 | 70.60 | 89.55 |
| Yelp | 87.44 | 93.20 | 87.44 | 96.20 |
| AG News | 89.31 | 93.54 | 85.85 | 96.05 |
| 20 NG | 60.75 | 60.75 | 60.75 | 94.22 |
| SST | 76.69 | 83.53* | 76.69 | 83.53 |
| Amzn | 69.91 | 88.07 | 57.36 | 90.38 |
| Muti30k | 22.94 | 36.61 | 22.94 | 66.29 |
| IWSLT14 | 29.07 | 32.98 | 29.07 | 67.36 |
| News It-Pt | 14.01 | 18.25 | 8.10 | 55.41 |
| News En-Nl | 9.60 | 18.59 | 9.60 | 62.90 |
| News En-Pt | 14.10 | 14.10 | 7.71 | 67.75 |

Table 2: Table 1 except with agreement defined by Kendall Tau. Section A.4

| Task | $\mathcal{A}(\alpha, \beta^{\text{uf}})$ | $\mathcal{A}(\beta^{\text{uf}}, \beta^{\text{px}})$ | $\mathcal{A}(\Delta, \beta^{\text{uf}})$ | $\hat{\mathcal{A}}$ |
|---|---|---|---|---|
| Muti30k | 8.68 | 27.54 | 4.24 | 0.00 |
| IWSLT14 | 8.64 | 22.56 | 2.72 | 0.00 |
| News It-Pt | 4.82 | 17.16 | 1.63 | 0.00 |
| News En-Nl | 4.53 | 20.35 | 2.08 | 0.00 |
| News En-Pt | 4.41 | 18.20 | 2.05 | 0.00 |
| Task | $\xi(\alpha, \beta^{\text{uf}})$ | $\xi(\beta^{\text{uf}}, \beta^{\text{px}})$ | $\xi(\Delta, \beta^{\text{uf}})$ | $\xi^*$ |
| Muti30k | 1.99 | 6.91 | 1.99 | 37.89 |
| IWSLT14 | 5.38 | 5.31 | 5.38 | 32.95 |
| News It-Pt | 0.09 | 0.55 | 0.04 | 24.71 |
| News En-Nl | 0.01 | 0.94 | 0.01 | 29.42 |
| News En-Pt | 0.01 | 0.22 | 0.01 | 37.04 |

Table 3: translation results from Table 2 except with performance measured by bleu rather than token accuracy. Section A.4

- Table 4 contains the same results as Table 1, except that results are now rounded to two decimal places.

- Table 6 consists of the same results as Table 1, except that the statistics is calculated over the training set rather than the validation set.

- Table 3, Table 5, and Table 7 contain the translation results from the above 3 mentioned tables respectively, except that $\hat{p}$ is defined as BLEU score rather than token accuracy, and hence the contextualized metric interpretation $\xi$ changes correspondingly.

## A.5 DATASET DESCRIPTION

We summarize the datasets that we use for classification and machine translation. See Table 8 for details on train/test splits and median sequence lengths for each dataset.

| Task | $\mathcal{A}(\alpha, \beta^{\mathrm{uf}})$ | $\mathcal{A}(\beta^{\mathrm{uf}}, \beta^{\mathrm{px}})$ | $\mathcal{A}(\Delta, \beta^{\mathrm{uf}})$ | $\mathcal{A}(\alpha, \beta)$ | $\hat{\mathcal{A}}$ |
|---|---|---|---|---|---|
| IMDB | 52.55 | 81.60 | 61.55 | 60.35 | 5.30 |
| Yelp | 17.55 | 75.38 | 58.90 | 35.00 | 5.80 |
| AG News | 39.24 | 55.13 | 43.08 | 48.13 | 6.20 |
| 20 NG | 65.08 | 41.33 | 64.82 | 63.07 | 5.11 |
| SST | 19.85 | 33.57 | 22.45 | 25.33 | 8.39 |
| Amzn | 52.02 | 76.78 | 49.68 | 62.13 | 5.80 |
| Muti30k | 31.02 | 34.43 | 27.06 | 48.78 | 7.11 |
| IWSLT14 | 35.75 | 39.09 | 27.69 | 55.25 | 6.52 |
| News It-Pt | 29.13 | 38.62 | 25.45 | 52.48 | 6.17 |
| News En-Nl | 35.53 | 41.72 | 29.15 | 60.15 | 6.36 |
| News En-Pt | 35.90 | 37.37 | 30.23 | 65.49 | 6.34 |
| Task | $\xi(\alpha, \beta^{\mathrm{uf}})$ | $\xi(\beta^{\mathrm{uf}}, \beta^{\mathrm{px}})$ | $\xi(\Delta, \beta^{\mathrm{uf}})$ | $\xi(\alpha, \beta)$ | $\xi^*$ |
| IMDB | 86.81 | 88.88 | 86.81 | 86.81 | 89.55 |
| Yelp | 90.39 | 95.22 | 95.31 | 93.59 | 96.20 |
| AG News | 93.54 | 96.05* | 94.32 | 94.50 | 96.05 |
| 20 NG | 91.16 | 60.75 | 84.57 | 84.57 | 94.22 |
| SST | 78.16 | 83.53* | 78.16 | 82.38 | 83.53 |
| Amzn | 82.48 | 90.38* | 82.48 | 88.07 | 90.38 |
| Muti30k | 43.45 | 43.45 | 43.45 | 48.58 | 66.29 |
| IWSLT14 | 35.82 | 35.82 | 32.98 | 44.09 | 67.36 |
| News It-Pt | 21.82 | 25.06 | 21.82 | 25.06 | 55.41 |
| News En-Nl | 18.59 | 23.21 | 18.59 | 26.79 | 62.90 |
| News En-Pt | 19.12 | 19.12 | 19.12 | 27.85 | 67.75 |

Table 4: Table 1 with 2 decimal results. Section A.4

| Task | $\mathcal{A}(\alpha, \beta^{\mathrm{uf}})$ | $\mathcal{A}(\beta^{\mathrm{uf}}, \beta^{\mathrm{px}})$ | $\mathcal{A}(\Delta, \beta^{\mathrm{uf}})$ | $\mathcal{A}(\alpha, \beta)$ | $\hat{\mathcal{A}}$ |
|---|---|---|---|---|---|
| Muti30k | 30.77 | 34.43 | 27.24 | 48.70 | 7.19 |
| IWSLT14 | 35.75 | 39.09 | 27.69 | 55.25 | 6.52 |
| News It-Pt | 29.13 | 38.62 | 25.45 | 52.48 | 6.17 |
| News En-Nl | 35.53 | 41.72 | 29.15 | 60.15 | 6.35 |
| News En-Pt | 35.77 | 37.37 | 30.37 | 64.94 | 6.34 |
| Task | $\xi(\alpha, \beta^{\mathrm{uf}})$ | $\xi(\beta^{\mathrm{uf}}, \beta^{\mathrm{px}})$ | $\xi(\Delta, \beta^{\mathrm{uf}})$ | $\xi(\alpha, \beta)$ | $\xi^*$ |
| Muti30k | 11.43 | 11.43 | 11.43 | 16.41 | 37.89 |
| IWSLT14 | 6.71 | 6.71 | 5.31 | 9.89 | 32.95 |
| News It-Pt | 1.29 | 2.16 | 1.29 | 2.16 | 24.71 |
| News En-Nl | 0.94 | 2.39 | 0.94 | 4.12 | 29.42 |
| News En-Pt | 0.74 | 0.74 | 0.74 | 4.28 | 37.04 |

Table 5: translation results from Table 4 except with performance measured by bleu rather than token accuracy. Section A.4

| Task | $\mathcal{A}(\alpha, \beta^{\text{uf}})$ | $\mathcal{A}(\beta^{\text{uf}}, \beta^{\text{px}})$ | $\mathcal{A}(\Delta, \beta^{\text{uf}})$ | $\mathcal{A}(\alpha, \beta)$ | $\hat{\mathcal{A}}$ |
|---|---|---|---|---|---|
| IMDB | 51.52 | 80.10 | 42.85 | 64.88 | 5.29 |
| Yelp | 11.15 | 76.12 | 55.50 | 37.63 | 5.85 |
| AG News | 36.97 | 53.95 | 43.11 | 46.89 | 6.17 |
| 20 NG | 72.36 | 38.69 | 71.73 | 69.47 | 5.32 |
| SST | 21.82 | 29.35 | 20.48 | 28.50 | 8.48 |
| Amzn | 51.95 | 77.18 | 40.15 | 61.78 | 5.91 |
| Muti30k | 32.89 | 34.67 | 28.36 | 56.39 | 7.21 |
| IWSLT14 | 36.61 | 38.95 | 28.37 | 57.71 | 6.52 |
| News It-Pt | 31.03 | 38.70 | 27.11 | 64.81 | 6.15 |
| News En-Nl | 37.86 | 41.91 | 31.11 | 67.68 | 6.39 |
| News En-Pt | 37.43 | 37.23 | 31.76 | 71.96 | 6.35 |
| Task | $\xi(\alpha, \beta^{\text{uf}})$ | $\xi(\beta^{\text{uf}}, \beta^{\text{px}})$ | $\xi(\Delta, \beta^{\text{uf}})$ | $\xi(\alpha, \beta)$ | $\xi^*$ |
| IMDB | 90.40 | 99.95* | 90.40 | 95.01 | 99.95 |
| Yelp | 75.61 | 96.54 | 96.19 | 94.44 | 98.22 |
| AG News | 93.57 | 98.42* | 94.63 | 95.54 | 98.42 |
| 20 NG | 100.00 | 65.40 | 100.00 | 100.0 | 100.00 |
| SST | 97.72 | 100.00* | 84.11 | 100.0* | 100.00 |
| Amzn | 87.96 | 99.58* | 80.98 | 91.09 | 99.58 |
| Muti30k | 43.27 | 43.27 | 43.27 | 51.97 | 80.76 |
| IWSLT14 | 35.94 | 35.94 | 35.94 | 44.18 | 71.18 |
| News It-Pt | 22.69 | 25.96 | 22.69 | 39.98 | 77.10 |
| News En-Nl | 18.85 | 23.56 | 18.85 | 40.09 | 74.49 |
| News En-Pt | 19.33 | 19.33 | 19.33 | 42.41 | 77.97 |

Table 6: Table 1 except with correlations and performance metrics taken over the training set instead of the validation set. Section A.4

| Task | $\mathcal{A}(\alpha, \beta^{\text{uf}})$ | $\mathcal{A}(\beta^{\text{uf}}, \beta^{\text{px}})$ | $\mathcal{A}(\Delta, \beta^{\text{uf}})$ | $\hat{\mathcal{A}}$ |
|---|---|---|---|---|
| Muti30k | 32.89 | 34.67 | 28.36 | 7.16 |
| IWSLT14 | 36.61 | 38.95 | 28.37 | 6.54 |
| News It-Pt | 31.03 | 38.70 | 27.11 | 6.17 |
| News En-Nl | 37.86 | 41.91 | 31.11 | 6.38 |
| News En-Pt | 37.43 | 37.23 | 31.76 | 6.37 |
| Task | $\xi(\alpha, \beta^{\text{uf}})$ | $\xi(\beta^{\text{uf}}, \beta^{\text{px}})$ | $\xi(\Delta, \beta^{\text{uf}})$ | $\xi^*$ |
| Muti30k | 11.87 | 11.87 | 11.87 | 52.28 |
| IWSLT14 | 6.82 | 6.82 | 6.82 | 36.23 |
| News It-Pt | 1.30 | 2.30 | 1.30 | 42.40 |
| News En-Nl | 1.11 | 2.29 | 1.11 | 39.40 |
| News En-Pt | 0.83 | 0.83 | 0.83 | 46.57 |

Table 7: translation results from Table Table 6 except with performance measured by bleu rather than token accuracy. Section A.4

**IMDB Sentiment Analysis** Maas et al. (2011) A sentiment analysis data set with 50,000 (25,000 train and 25,000 test) IMDB movie reviews and their corresponding positive or negative sentiment.

**AG News Corpus** Zhang et al. (2015) 120,000 news articles and their corresponding topic (world, sports, business, or science/tech). We classify between the world and business articles.

**20 Newsgroups** [4] A news data set containing around 18,000 newsgroups articles split between 20 different labeled categories. We classify between baseball and hocky articles.

**Stanford Sentiment Treebank** Socher et al. (2013) A data set for classifying the sentiment of movie reviews, labeled on a scale from 1 (negative) to 5 (positive). We remove all movies labeled as 3, and classify between 4 or 5 and 1 or 2.

**Multi Domain Sentiment Data set** [5] Approximately 40,000 Amazon reviews from various product categories labeled with a corresponding positive or negative label. Since some of the sequences are particularly long, we only use sequences of length less than 400 words.

**Yelp Open Data Set** [6] 20,000 Yelp reviews and their corresponding star rating from 1 to 5. We classify between reviews with rating $\leq 2$ and $\geq 4$.

**Multi-30k** Elliott et al. (2016) English to German translation. The data is from translation image captions.

**IWSLT'14** Cettolo et al. (2015) German to English translation. The data is from translated TED talk transcriptions.

**News Commentary v14** Cettolo et al. (2015) A collection of translation news commentary datasets in different languages from WMT19 [7]. We use the following translation splits: English-Dutch (En-Nl), English-Portuguese (En-Pt), and Italian-Portuguese (It-Pt). In pre-processing for this dataset, we removed all purely numerical examples.

## A.6 $\alpha$ Fails When $\beta$ is Frozen

For each classification task we initialize a random model and freeze all parameters except for the attention layer (frozen $\beta$ model). We then compute the correlation between this trained attention (defined as $\alpha^{\mathrm{fr}}$) and the normal attention $\alpha$. Table 9 reports this correlation at the iteration where $\alpha^{\mathrm{fr}}$ is most correlated with $\alpha$ on the validation set. As shown in Table 9, the left column is consistently lower than the right column. This indicates that the model can learn output relevance without attention, but not vice versa.

## A.7 Training $\beta^{\mathrm{uf}}$

We find that $\mathcal{A}(\alpha, \beta^{\mathrm{uf}}(\tau))$ first increases and then decreases as training proceeds (i.e. $\tau$ increases), so we chose the maximum agreement to report in Table 1 over the course of training. Since this trend is consistent across all datasets, our choice minimally inflates the agreement measure, and is comparable to the practice of reporting dev set results. As discussed in Section 6.1, training under uniform attention for too long might bring unintuitive results,

## A.8 Model and Training Details

**Classification** Our model uses dimension 300 GloVe-6B pre-trained embeddings to initialize the token embeddings where they aligned with our vocabulary. The sequences are encoded with a 1 layer bidirectional LSTM of dimension 256. The rest of the model, including the attention mechanism, is exactly as described in 2.4. Our model has 1,274,882 parameters excluding embeddings. Since each classification set has a different vocab size each model has a slightly different parameter count when considering embeddings: 19,376,282 for IMDB, 10,594,382 for AG News, 5,021,282 for 20

---

[4]http://qwone.com/ jason/20Newsgroups/

[5]https://www.cs.jhu.edu/ mdredze/datasets/sentiment/

[6]https://www.yelp.com/dataset

[7]http://www.statmt.org/wmt19/translation-task.html

| Data | median train seq len | train # | Data | median val seq len | val # |
|---|---|---|---|---|---|
| IMDB | 181 | 25000 | IMDB | 178 | 4000 |
| AG News | 40 | 60000 | AG News | 40 | 3800 |
| NewsG | 183 | 1197 | NewsG | 207 | 796 |
| SST | 16 | 5130 | SST | 17 | 1421 |
| Amzn | 71 | 32514 | Amzn | 72 | 4000 |
| Yelp | 74 | 88821 | Yelp | 74 | 4000 |
| IWSLT14 | (23 src, 24 trg) | 160240 | IWSLT14 | (22 src, 23 trg) | 7284 |
| Multi-30k | (14 src, 14 trg) | 29000 | Multi-30k | (15 src, 14 trg) | 1014 |
| News-en-nl | (30 src, 34 trg) | 52070 | News-en-nl | (30 src, 34 trg) | 5786 |
| News-en-pt | (31 src, 35 trg) | 48538 | News-en-pt | (31 src, 35 trg) | 5394 |
| News-it-pt | (36 src, 35 trg) | 21572 | News-it-pt | (36 src, 36 trg) | 2397 |

| Data | vocab size |
|---|---|
| IMDB | 60338 |
| AG News | 31065 |
| NewsG | 31065 |
| SST | 11022 |
| Amzn | 37110 |
| Yelp | 41368 |
| IWSLT14 | 8000 |
| Multi-30k | 8000 |
| News-en-nl | 8000 |
| News-en-pt | 8000 |
| News-it-pt | 8000 |

Table 8: statistics for each dataset. Median sequence length in the training set and train set size. Note: src refers to the input "source" sequence, and trg refers to the output "target" sequence. Section A.5

| Dataset | $\mathcal{A}(\alpha, \alpha^{\mathrm{fr}})$ | $\mathcal{A}(\alpha, \beta^{\mathrm{uf}})$ |
|---|---|---|
| IMDB | 9 | 53 |
| AG News | 17 | 39 |
| 20 NG | 19 | 65 |
| SST | 14 | 20 |
| Amzn | 15 | 52 |
| Yelp | 8 | 18 |

Table 9: We report the correlation between $\alpha^{\mathrm{fr}}$ and $\alpha$ on classification datasets, and compare it against $\mathcal{A}(\alpha, \beta^{\mathrm{uf}})$, the same column defined in Table 1. Section A.6

Newsgroups, 4,581,482 for SST, 13,685,282 for Yelp, 12,407,882 for Amazon, and 2,682,182 for SMS.

**Translation** We use a a bidirectional two layer bi-LSTM of dimension 256 to encode the source and the use last hidden state $h_L$ as the first hidden state of the decoder. The attention and outputs are then calculated as described in 2. The learn-able neural network before the outputs that is mentioned in Section 2, is a 1 hidden layer model with ReLU non-linearity. The hidden layer is dimension 256. Our model contains 6,132,544 parameters excluding embeddings and 8,180,544 including embeddings on all datasets.

**Permutation Copying** We use single directional single layer LSTM with hidden dimension 256 for both the encoder and the decoder.

**Classification Procedure** For all classification datasets we used a batch size of 32. We trained for 4000 iterations on each dataset. For each dataset we train on the pre-defined training set if the dataset has one. Additionally, if a dataset had a predefined test set, we randomly sample at most 4000 examples from this test set for validation. Specific dataset split sizes are given in Table 8.

**Classification Evaluation**    We evaluated each model at steps 0, 10, 50, 100, 150, 200, 250, and then every 250 iterations after that.

**Classification Tokenization**    We tokenized the data at the word level. We mapped all words occurring less than 3 times in the training set to <unk>. For 20 Newsgroups and AG News we mapped all non-single digit integer "words" to <unk>. For 20 Newsgroups we also split words with the "_" character.

**Classification Training**    We trained all classification models on a single GPU. Some datasets took slightly longer to train than others (largely depending on average sequence length), but each train took at most 45 minutes.

**Translation Hyper Parameters**    For translation all hidden states in the model are dimension 256. We use the sequence to sequence architecture described above. The LSTMs used dropout 0.5.

**Translation Procedure**    For all translation tasks we used batch size 16 when training. For IWSLT'14 and Multi-30k we used the provided dataset splits. For the News Commentary v14 datasets we did a 90-10 split of the data for training and validation respectively.

**Translation Evaluation**    We evaluated each model at steps 0, 50, 100, 500, 1000, 1500, and then every 2000 iterations after that.

**Translation Training**    We trained all translation models on a single GPU. IWSLT'14, and the News Commentary datasets took approximately 5-6 hours to train, and multi-30k took closer to 1 hour to train.

**Translation Tokenization**    We tokenized both translation datasets using the Sentence-Piece tokenizer trained on the corresponding train set to a vocab size of 8,000. We used a single tokenization for source and target tokens. And accordingly also used the same matrix of embeddings for target and source sequences.

### A.9    A Note On SMS Dataset

In addition to the classification datasets reported in the tables, we also ran experiments on the SMS Spam Collection V.1 dataset [8]. The attention learned from this dataset was very high variance, and so two different random seeds would consistently produce attentions that did not correlate much. The dataset itself was also a bit of an outlier; it had shorter sequence lengths than any of the other datasets (median sequence length 13 on train and validation set), it also had the smallest training set out of all our datasets (3500 examples), and it had by far the smallest vocab (4691 unique tokens). We decided not to include this dataset in the main paper due to these unusual results and leave further exploration to future works.

### A.10    Logistic Regression Proxy Model

Our proxy model can be shown to be equivalent to a bag-of-words logistic regression model in the classification case. Specifically, we define a bag-of-words logistic regression model to be:

$$\forall t, p_t = \sigma(\beta^{\log} x). \tag{52}$$

where $x \in \mathbb{R}^{|\mathcal{I}|}$, $\beta^{\log} \in \mathbb{R}^{|\mathcal{O}| \times |\mathcal{I}|}$, and $\sigma$ is the softmax function. The entries in $x$ are the number of times each word occurs in the input sequence, normalized by the sequence length. and $\beta^{\log}$ is learned. This is equivalent to:

---

[8]http://www.dt.fee.unicamp.br/ tiago/smsspamcollection/

| Task | $\mathcal{A}(\beta^{\mathrm{uf}}, \beta^{\mathrm{px}})$ | $\mathcal{A}(\beta^{\mathrm{uf}}, \beta^{\mathrm{log}})$ |
|---|---|---|
| IMDB | 0.81 | 0.84 |
| Yelp | 0.74 | 0.76 |
| AG News | 0.57 | 0.58 |
| 20 NG | 0.40 | 0.45 |
| SST | 0.39 | 0.46 |
| Amzn | 0.53 | 0.60 |

Table 10: we report $\mathcal{A}(\beta^{\mathrm{uf}}, \beta^{\mathrm{log}})$ to demonstrate its effective equivalence to $\mathcal{A}(\beta^{\mathrm{uf}}, \beta^{\mathrm{px}})$. These values are not exactly the same due to differences in regularization strategies.

$$\forall t, p_t = \sigma(\frac{1}{L} \sum_{l=1}^{L} \beta_{x_l}^{\mathrm{log}}).$$ (53)

Here $\beta_i^{\mathrm{log}}$ indicates the $i$th column of $\beta^{\mathrm{log}}$; these are the entries in $\beta^{\mathrm{log}}$ corresponding to predictions for the $i$th word in the vocab. Now it is easy to arrive at the equivalence between logistic regression and our proxy model. If we restrict the rank of $\beta^{\mathrm{log}}$ to be at most $\min(d, |O|, |I|)$ by factoring it as $\beta^{\mathrm{log}} = WE$ where $W \in \mathbb{R}^{|\mathcal{O}| \times d}$ and $E \in \mathbb{R}^{d \times |\mathcal{I}|}$, then the logistic regression looks like:

$$\forall t, p_t = \sigma(\frac{1}{L} \sum_{l=1}^{L} WE_{x_l}),$$ (54)

which is equivalent to our proxy model:

$$\forall t, p_t = \sigma(\frac{1}{L} \sum_{l=1}^{L} We_{x_l}).$$ (55)

Since $d = 256$ for the proxy model, which is larger than $|O| = 2$ in the classification case, the proxy model is not rank limited and is hence fully equivalent to the logistic regression model. Therefore the $\beta^{\mathrm{px}}$ can be interpreted as "keywords" in the same way that the logistic regression weights can.

To empirically verify this equivalence, we trained a logistic regression model with $\ell 2$ regularization on each of our classification datasets. To pick the optimal regularization level, we did a sweep of regularization coefficients across ten orders of magnitude and picked the one with the best validation accuracy. We report results for $\mathcal{A}(\beta^{\mathrm{uf}}, \beta^{\mathrm{log}})$ in comparison to $\mathcal{A}(\beta^{\mathrm{uf}}, \beta^{\mathrm{px}})$ in Table 10 [9].

Note that these numbers are similar but not exactly equivalent. The reason is that the proxy model did not use $\ell 2$ regularization, while logistic regression did.

---

[9]These numbers were obtained from a retrain of all the models in the main table, so for instance, the LSTM model used to produce $\beta^{\mathrm{uf}}$ might not be exactly the same as the one used for the results in all the other tables due to random seed difference.

