# OpenReview forum: "Approximating How Single Head Attention Learns"
_ICLR.cc/2023/Conference — Submitted to ICLR 2023_

### Official Review · Reviewer_AtcN · 2022-10-24

**Confidence:** 3
**Correctness:** 2
**Technical Novelty And Significance:** 2
**Empirical Novelty And Significance:** 2
**Recommendation:** 3

**Clarity, Quality, Novelty And Reproducibility:**

The quality of this paper would increase if it more clearly demonstrated that the proposed framework matches learning dynamics on non-toy tasks. The paper is mostly clear. To my knowledge, the work is original, but the contributions are arguably limited. The paper should be mostly reproducible with the attached code.

**Strength And Weaknesses:**

Strengths

The experiment showing that copying sequences is a difficult task under some constraints is insightful.

The paper shows how multi-head attention can improve learning dynamics.

Weaknesses

The paper presents a plausible 2-stage learning approach for attention, but it doesn't really show that this happens in practice. For example, we could monitor the entropies of the output distributions and of the attention weights during training.

The simplifying assumptions are quite strong. Under the proposed framework, we may not be able to explain phenomena such as Figure 9 in [1], where attention does not necessarily match work alignment.

[1] Koehn and Knowles. Six Challenges for Neural Machine Translation. First Workshop on Neural Machine Translation. 2017

**Summary Of The Paper:**

This paper presents a framework explaining how attention might be learned. First, the model would get the knowledge to translate individual words (KTIW) based on word co-occurences, which can be learned if the attention weights are uniform. KTIW then drives the learning of the attention mechanism.

**Summary Of The Review:**

The paper proposes an interesting and plausible framework of how attention is learnt. However, it does not clearly show that the 2 stages of learning happen in practice (is it really sequential?). Some of the assumptions may be too restrictive.

---

### Official Review · Reviewer_H7cW · 2022-10-26

**Confidence:** 2
**Correctness:** 2
**Technical Novelty And Significance:** 2
**Empirical Novelty And Significance:** 2
**Recommendation:** 3

**Clarity, Quality, Novelty And Reproducibility:**

The text is quite hard to parse, and the introduction doesn’t do a great job at justifying the work. The paper would benefit from some work to improve the logical flow in the presentation of ideas and arguments.

The title suggests that the paper is broadly about understanding attention mechanisms but really the paper focuses on machine translation and seq2seq models. It would be helpful for the reader if this information was made explicit in the title and abstract.


**Strength And Weaknesses:**

*Strengths:*
- Aiming for a theoretical understanding of learning dynamics and attention is an admirable goal. The high level topic would be of interest to many in the community.

*Weaknesses:*
-  The major weakness of this work is that I found the text and presentation of results really difficult to parse. It is also difficult to establish what exactly each experiment contributes to the logic of the paper, why they were designed in that way, or the takeaways from each expreriment, which made it quite a difficult read.

- What are the implications of the results in this paper? I think I am missing the major takeaway from this work. It is not clear to me how defining this proxy for machine translation is useful for interpretability and understanding machine translation learning dynamics.

- The KTIW proxy seems to assume a 1:1 ratio of input to output words for translation, which surely cannot hold true in the general MT case? Perhaps I am missing something here.

- If I understand correctly, the logic is section 5.3 seems to be circular. The paper defines the initialisation of an attention head as ‘good’ if training with that single head converges (on a task that requires only a single attention head). The paper then constructs multi-head attention scenarios with all bad initialisations, all random, or only-one-good head initialisation. They present the result that if all initialisations are bad, they model does not learn, but if at least on is ‘good’ then the model will converge. As the task is learnable with a single attention head, this logic seems circular and trivially true. It is not clear to me what it adds to the paper and seems off-topic.


**Summary Of The Paper:**

This paper investigates how seq2seq attention weights are learned in a machine translation setting. The paper defines a proxy to measure a model’s ability to translate individual words, which they shorten to KTIW. The paper claims that this measurable quantity is a driver for the model to learn to attend well. The evidence presented in favour of this hypothesis is that KTIW can be learned (it is similar to learned attention weights) when attention weights are frozen and  uniform, but that when KTIW is not learned, neither are the attention weights. This suggests a sort of necessary condition on KTIW for attention learning, which the paper claims is a causal driver for attention learning.

The paper suggests that it is therefore possible to reduce the problem of understanding attention learning to the problem of understanding KTIW (which does not seem to be entirely true, as KTIW may be necessary but not sufficient condition for learning of attention weights).
The paper thus proposes a proxy model for approximating KTIW learning (that is, a proxy for the proxy for attention weight learning), and verifies this. The paper claims that attention weights are learned in two stages: first KTIW is learned through word co-occurence statistics, and second, a learned KTIW drives the learning of attention.


**Summary Of The Review:**

This paper investigates how seq2seq attention weights are learned in a machine translation setting. The paper defines a proxy to measure a model’s ability to translate individual words, which they shorten to KTIW. I found this paper very hard to follow, as the writing, justifications and logical flow of the word hid the scientific contributions. The paper would benefit from significant clarifying and removal of experiments (e.g. section 5.3) that detract from the points the authors wish the readership to takeaway - this might enable more readers to see the scientific contributions of the paper to the wider literature more clearly.

---

### Official Review · Reviewer_8y4q · 2022-10-27

**Confidence:** 3
**Clarity, Quality, Novelty And Reproducibility:** 1. Why a symmetric KL between the dis…
**Correctness:** 2
**Technical Novelty And Significance:** 2
**Empirical Novelty And Significance:** 2
**Recommendation:** 3

**Strength And Weaknesses:**

Strength:
1. The paper did some rigorous analysis of the attention training mechanism and drew a connection to the word alignment of the classical models.
2. There is a connection to interpretable classification for the mechanism described in the paper.

Weakness:
1. The applicability of such an understanding is now clear.
2. The analysis has a lot of underlying assumption (almost all of them are pointed out by the authors) which does not usually hold.

**Summary Of The Paper:**

The paper aims to understand the training dynamics of the Attention mechanism through lexical prob $\beta$ and its learning proxy model.


**Summary Of The Review:**

All the observations made are interesting, and the intuitive connections are also very fascinating. However, I still fear that these analyses might not have many practical applications and even won't hold in many different learning dynamics.

---

### Decision · Program_Chairs · 2023-01-20

**Decision:**

Reject

**Justification For Why Not Higher Score:**

The authors did not respond to reviewers' concerns, during the response period

**Justification For Why Not Lower Score:**

N/A

**Metareview: Summary, Strengths And Weaknesses:**

The paper attempts to explain the training dynamics of the attention mechanism in a seq2seq setting. The central conjecture of this work is that the attention mechanism first learns word-level translations (a phenomenon termed “Knowledge to Translate Individual Words” or KTIW) and such KTIW then drives the learning of attention. All the reviewers concur that this work aims to tackle a significant problem. However, there are major outstanding concerns around clarity (reviewers H7cW and AtcN), lack of empirical evidence (reviewers H7cW, AtcN, and 8y4q), and overly restrictive assumptions (reviewers 8y4q and AtcN). Unfortunately, the authors did not address the reviewers’ concerns during the response period. Thus, the paper is recommended for rejection.

**Summary Of Ac-Reviewer Meeting:**

N/A